

# DEEP Picker1D and Voigt Fitter1D: A versatile tool set for the automated quantitative spectral deconvolution of complex 1D NMR spectra

Da-Wei Li,[1*] Lei Bruschweiler-Li,[1] Alexandar L. Hansen,[1] and Rafael Brüschweiler[1,2,3*]

[1]Campus Chemical Instrument Center, The Ohio State University, Columbus, Ohio 43210, USA

[2]Department of Chemistry and Biochemistry, The Ohio State University, Columbus, Ohio 43210, USA

[3]Department of Biological Chemistry and Pharmacology, The Ohio State University, Columbus, Ohio 43210, USA

[*]To whom correspondence should be addressed:

Da-Wei Li, Ph.D.,  E-mail: lidawei@gmail.com

Rafael Brüschweiler, Ph.D.,  E-mail: bruschweiler.1@osu.edu

**Abstract**

The quantitative deconvolution of 1D NMR spectra into individual resonances or peaks is a key step in many modern NMR workflows as it critically affects downstream analysis and interpretation. Depending on the complexity of the NMR spectrum, spectral deconvolution can

be a notable challenging. Based on the recent deep neural network DEEP Picker and Voigt Fitter for 2D NMR spectral deconvolution, we present here an accurate, fully automated solution for 1D NMR spectral analysis, including peak picking, fitting, and reconstruction. The method is demonstrated for complex 1D solution NMR spectra showing excellent performance also for spectral regions with multiple strong overlaps and a large dynamic range whose analysis is

challenging for current computational methods. The new tool will help streamline 1D NMR spectral analysis for a wide range of applications and expand their reach toward ever more complex molecular systems and their mixtures.

*Keywords:* NMR spectral analysis, complex NMR spectra, spectral overlap, machine learning,

deep neural network, peak picking, DEEP Picker, peak fitting, lineshape fitting, spectral reconstruction.

**MAGNETIC RESONANCE**
Open Access Discussions

## 1. Introduction

One of the major strengths of nuclear magnetic resonance (NMR) spectroscopy is its broad
applicability to a vast range of molecular systems in solution or in the solid state. Because the
nuclei of many atoms in molecules are NMR-active, such as hydrogen atoms, the information
content of NMR spectra is uniquely rich allowing studies of molecular composition, interactions,
structure and dynamics at atomic detail. Due to its quantitative nature, NMR is also highly
suitable for the analysis of molecular mixtures for component identification and quantification
with application in metabolomics (Markley et al., 2017) and for monitoring of industrial
chemical and biochemical processes (Wang et al., 2021).

Despite enormous methodological progress made over many decades of NMR research
that have resulted in a vast collection of different NMR experiments, in many NMR facilities the
most popular choice remains the standard one-dimensional (1D) [1]H proton NMR experiment.
This is the result of several factors, such as good sensitivity, short measurement time (potentially
associated with a low user fee), straightforward processing, and easy and dependable
implementation on different types of NMR spectrometers. However, due to the richness of the
resulting [1]H NMR spectrum in many samples, it is prone to various amounts of spectral overlaps,
which is the overlap of two or more resonances, rendering the identification and quantification of
the underlying resonances challenging (Giraudeau, 2017).

Because the first step of the analysis of almost every NMR spectrum consists of the
identification of the individual resonances, spectral crowding often makes the process incomplete,
ambiguous or even impossible. For many years, spectral analysis is being routinely assisted by
computer software to perform useful tasks like peak-picking and peak integration thereby
speeding up the analysis process by supporting human experts during this process (Nelson and

Brown, 1989; Martin, 1994; Cobas et al., 2013). A number of commercial general purpose software packages are available for the analysis of 1D $^1$H NMR spectra such as the ACD/NMR workbook suite (https://www.acdlabs.com/), the AMIX software (https://www.bruker.com), the Chenomx NMR suite (https://www.chenomx.com), and MNova NMR

(https://www.mestrelab.com). Recent developments in NMR-based metabolomics, which oftentimes involve highly complex $^1$H NMR spectra, has led to a proliferation of academic software for the (semi-)automated analysis of such spectra, including MetaboLab (Ludwig and Gunther, 2011), BATMAN (Hao et al., 2014), Bayesil (Ravanbakhsh et al., 2015), AQuA (Rohnisch et al., 2018), ASICS (Lefort et al., 2019), rDolphin (Canueto et al., 2018) and

MetaboDecon1D (Hackl et al., 2021). Some of these programs are suitable for untargeted compound identification whereas others only map those spectral features that are contained in a pre-defined metabolite spectral database.

For a fully quantitative spectral analysis, numerical lineshape fitting has become the method of choice using a parametric representation of each resonance in the spectrum

(Higinbotham and Marshall, 2001; Smith, 2017; Sokolenko et al., 2019). Commonly used lineshapes are Lorentzian, Gaussian, and Voigt profiles that may explicitly include truncation or apodization effects, such as sinc wiggles (Dudley et al., 2020). Because essentially all fitting software rely on a local non-linear least squares minimization between the model and the experimental spectrum, such as a Levenberg-Marquardt minimizer, accurate line position and

linewidth for each resonance as input parameters is of paramount importance. Because such information is hard to obtain by automated computational approaches alone, lineshape fitting often requires significant interactive intervention by a human expert. This applies in particular to spectral regions with significant peak overlap manifested, for example, by one or several

shoulder peaks and a large dynamic range. Although sophisticated mathematical peak picking

algorithms have been developed that identify realistic peak positions (Cobas et al., 2013), they

work best for well-resolved peaks or peaks with moderate overlap, but tend to fail in the case of

strong overlaps and overlaps involving three or more peaks.

Recent applications of machine-learning methods, in particular of deep neural networks

(DNN), have shown qualitative progress in the ability to deconvolute complex multidimensional

NMR spectra (Li et al., 2022b). In the case of "DEEP Picker", training was exclusively based on

a library containing 5000 synthetic 1D test spectra consisting of 3 to 9 individual Voigt-shaped

peaks with random amplitudes and positions amounting to a collection of training spectra with a

wide range of spectral overlap (Li et al., 2021). The algorithm was then generalized to two-

dimensional (2D) NMR spectra as encountered in many protein NMR and metabolomics

applications.

In the present work, we introduce DEEP Picker for untargeted applications to complex

1D NMR spectra, including complex biological mixtures. The deconvolution power of "DEEP

Picker1D" is demonstrated for spectra with various amounts of overlap and how it can be paired

with the non-linear least squares fitting software "Voigt Fitter1D" for a fully quantitative

deconvolution of the input spectra. The computer codes of DEEP Picker1D and Voigt Fitter1D

are made publicly available.


**MAGNETIC RESONANCE**
Discussions

### Materials and Methods

### Sample Preparation

**Glucose sample.** 2 mM glucose (from Sigma-Aldrich) was prepared in $D_2O$ before 600 μL were transferred to a 5 mm NMR tube for NMR data collection.

**Mouse urine sample.** Frozen mouse urine sample was thawed on ice. An aliquot of 178 μl mouse urine was mixed with 20 μl sodium phosphate buffer (500 mM) in $D_2O$ and 2 μl DSS (4,4-dimethyl-4-silapentane-1-sulfonic acid from 10 mM stock solution prepared in $D_2O$) with a final pH of 7.4. 200 μl of the final sample was transferred to a 3 mm NMR tube for NMR data collection.


### NMR Experiments and Processing

All NMR spectra were collected at 298 K on Bruker AVANCE III HD 850 MHz spectrometers equipped with a cryogenically cooled TCI probe. A 1D [1]H NOESY glucose spectrum was recorded with a total of 32768 complex data points and 64 scans. Relaxation delay between 100 consecutive scans was 12 s, the spectral width was 13 ppm, and the transmitter frequency offset was set to 4.7 ppm. NMR data was zero-filled four-fold, apodized using a cosine squared window function, Fourier-transformed, and phase-corrected using Bruker Topspin 4 software.

1D [1]H mouse urine spectrum was recorded with the Bruker standard pulse sequence "zgesgppe" with a total of 53190 complex data points and 64 scans. The relaxation delay 105 between consecutive scans was 4 s. The spectral width was 25 ppm with the transmitter frequency offset set to 4.7 ppm. The NMR free induction decay was zero-filled two-fold,

apodized using a 2π-Kaiser window function, Fourier-transformed, and phase-corrected using NMRPipe (Delaglio et al., 1995).

A 2D $^{13}$C-$^{1}$H high resolution HSQC spectrum of mouse urine was recorded with Bruker
pulse program "hsqcetgpsisp2.2", 3072 total complex data points in the $^{1}$H $t_2$ dimension and 512 total complex points in the $^{13}$C $t_1$ dimension were recorded. For each $t_1$ increment 16 scans were recorded and the relaxation delay between consecutive scans was set to 1.5 s. The spectral widths along the $^{1}$H and $^{13}$C dimensions were 18 ppm and 185 ppm, respectively. The transmitter frequency offsets were 4.7 ppm and 82.5 ppm, respectively. NMR data was zero-filled eight-fold
in both dimensions, apodized using a 2π-Kaiser window function, Fourier-transformed, and phase-corrected using NMRPipe (Delaglio et al., 1995).

**Deep neural network DEEP Picker1D and Voigt Fitter**

DEEP Picker1D is a deep neural network that was trained on a library of 5000 synthetic 1D
NMR spectra containing between 3 and 9 peaks with Voigt lineshape and variable amounts of overlaps (Li et al., 2021). In the original work, DEEP Picker was specifically adapted for the analysis of 2D NMR spectra and subsequently combined with the Voigt Fitter software for the quantitative analysis of 2D NMR metabolomics spectra either as standalone software or incorporated in the public web server COLMARq (Li et al., 2022a). Briefly, DEEP Picker1D is a
convolutional neural network, which was trained using TensorFlow v1.3 (Abadi et al., 2016), taking an 1D spectrum as input. It contains 7 hidden convolutional layers, 1 hidden max-pooling layer, and two parallel output layers with a total of 8037 trainable parameters. A convolutional output classifier layer with SoftMax activation classifies every input data point by

assigning an individual score for three peak classes (main peaks = class 2, shoulder peaks = class

1, no peak = 0). The class with the maximal score is then chosen as the predicted class with the

numerical score as a quantitative measure of confidence of the predicted class for each data point

of the input spectrum. For any data point predicted to be a peak (class 2 or 1), DEEP Picker1D

also predicts the sub-pixel peak position relative to the on-grid points, peak amplitude, peak

width, and the Lorentzian vs. Gaussian components to the Voigt shape using a convolutional

output regressor layer. Although DEEP Picker1D is a rather accurate predictor of peak

parameters in its own right, these values can be further refined by the Voigt Fitter1D software by

performing a non-linear least square fit of the original input 1D spectrum in terms of Voigt peak

shapes using the DEEP Picker1D output peak parameters as input. Voigt Fitter1D is essentially a

1D version of the 2D Voigt Fitter software published previously (Li et al., 2022a). DEEP

Picker1D paired with Voigt Fitter1D results in a fully quantitative representation of the input 1D

NMR spectrum in terms of a finite set of 1D Voigt-shaped peaks.

The input spectrum for DEEP Picker1D needs to be pre-processed in standard fashion,

including phase correction, baseline correction, zero filling, apodization and Fourier

transformation. DEEP Picker1D contains two models whereby model 1 (model 2) has optimal

performance when the digital resolution is sufficiently high around 12 (8) points per peak (PPP).

Deep Picker1D performs best for peaks with a moderate to high signal-to-noise ratio (S/N) and

lineshapes that closely follow Voigt profiles with a S/N > 10 where the noise level is defined as

the standard deviation of the spectrum in a peak-free region. In the presence of significant

amounts of noise, nonnegligible line shape distortion, such as those caused by temperature

fluctuations or suboptimal shimming during data collection, Deep Picker1D may pick some false

peaks, for example, by interpreting lineshape distortions as shoulder peaks. Voigt Fitter1D has

built-in tools to remove spectral features from its peak list when one of the following situations occurs: (i) a fitted peak is too wide, i.e., the peak width is larger than the fitting region, or it becomes too narrow, i.e. the peak width is less than 1 point; (ii) a fitted peak strongly overlaps

with another peak so that merging of two peaks into a single peak causes a minimal change of the fitting error. Deep Picker1D and Voigt Fitter1D together provide a self-sufficient spectral analysis tool set for the complete deconvolution of 1D spectra into individual peaks. Peak parameters, such as peak position, peak height and peak volume can then be directly used for downstream analysis, such as compound identification and quantitative NMR applications

(qNMR). Because error estimation is an important part of any quantitative data analysis, Monte Carlo-based error propagation is implemented in Voigt Fitter1D as an option. It performs repetitive fitting of the reconstructed spectrum after adding random noise with the same standard deviation as that of the experimental input spectrum for each round of fitting. The output from this error estimation procedure contains the fitting parameters from each round from which the

uncertainty of each peak parameter is obtained.


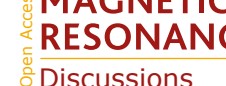

## 4. Results

DEEP Picker1D and Voigt Fitter1D performance is first demonstrated for glucose in $D_2O$

(**Figure 1**).

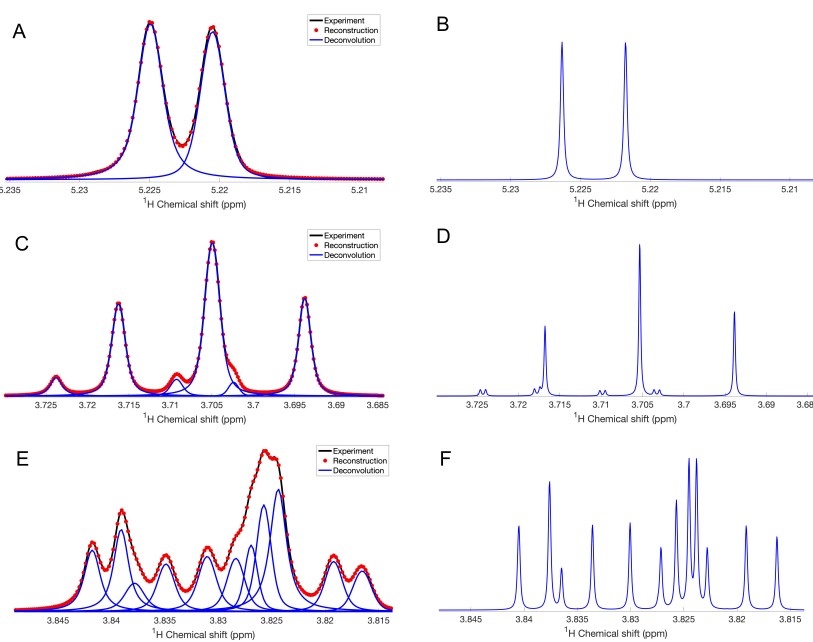

**Figure 1.** Demonstration of DEEP Picker and Voigt Fitter 1D for selected regions of 1D [1]H spectrum of glucose. (A), (C), (E): Experimental and reconstructed spectra are depicted in black lines and red dots, respectively. Deconvoluted individual peaks are depicted as blue lines. (B), (D), (F): Simulated spectra including strong coupling effects based on chemical shift and scalar-coupling spin Hamiltonian with parameters taken from GISSMO website at the same $B_0$ field strength (850 MHz [1]H frequency) as in the experiments. Transverse $R_2$ relaxation rates were uniformly set to a low value of 0.6 s[-1] to obtain a very high-resolution spectrum for better comparison with DEEP Picker. Pairs of panels (A, B), (C, D), (E, F) show the same 1D spectral regions. DEEP Picker and Voigt Fitter 1D correctly deconvoluted the experimental spectra for both simple regions (A) and more complex regions (C and E). Few peaks cannot be deconvoluted because of strong spectral overlap, such as the small peak around 3.717 ppm in (D) and the peak around 3.823 ppm in (F). The deconvolution by DEEP Picker1D was performed with model 2 with a PPP of 9.

Because glucose populates two non-equivalent isomers α-glucose and β-glucose with different relative populations that interconvert on a slow time scale and displays strong coupling effects even at high magnetic field, the deconvolution of its 1D $^1$H NMR spectrum is notoriously difficult. **Figure 1** shows selected regions of the 1D $^1$H NMR glucose spectrum with variable amounts of peak overlap. The experimental spectra (black) along with the deconvolution results (blue) are shown in the left column (Panels A, C, E). The right column (Panels B, D, F) shows the corresponding spectral regions derived from quantum-mechanical spin simulations using chemical shifts and scalar J-couplings obtained from the GISSMO library (Dashti et al., 2018). An artificially slow, uniform transverse $R_2$ relaxation rate of 0.6 s$^{-1}$ was applied to the simulated free induction decays (FID) so that after Fourier transformation, the resulting spectrum has sharp lines for easy recognition of the individual peaks and for the comparison with the automated deconvolution results. **Figure 1A** starts out with a symmetric doublet centered at 5.223 ppm, which is accurately picked and fitted by DEEP Picker1D and Voigt Fitter1D in agreement with the simulation results in **Figure 1B**. **Figure 1C,D** shows a triplet centered around 3.705 ppm whereby the strong central peak overlaps with two much smaller peaks on each side, which are correctly picked and fitted. According to the simulation, there is another small peak around 3.718 ppm, which however strongly overlaps with a much stronger peak at 3.716 ppm and therefore could not be identified by DEEP Picker1D. This small peak also cannot be discerned by visual inspection (note that the small J-splitting of the small peak in the simulated spectrum of **Figure 1D** are not resolved in the experimental spectrum of **Figure 1C**). The most complex region of the glucose spectrum (3.81 – 3.85 ppm) is depicted in **Figure 1E** along with its deconvolution, which is in very good agreement with the simulated peaks (**Figure 1F**). The neural network does



a remarkable job in identifying the small peak at 3.838 ppm, which only gives rise to a very faint

shoulder peak of its down-field shifted larger neighbor. The broad, somewhat oddly shaped

spectral feature from 3.82 to 3.83 ppm in the experiment is deconvoluted into 4 individual peaks

whereby the small peak found in the simulation at 3.823 ppm was not deconvoluted by DEEP

Picker1D because it overlaps too closely with the main peak at 3.824 ppm. This is consistent

with the general rule that two peaks whose positions differ within their linewidths are hard to

deconvolute, especially when their amplitudes significantly differ from each other.

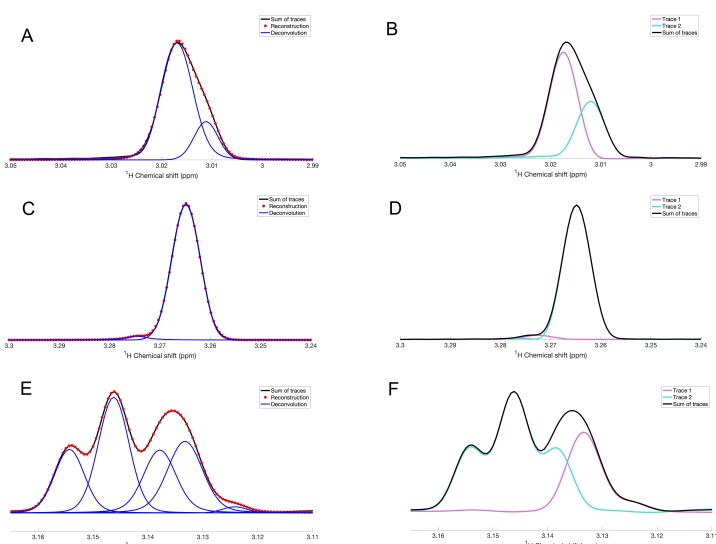

**Figure 2.** Application of DEEP Picker and Voigt Fitter 1D to selected regions of 1D spectra, which were generated by adding two selected traces along direct $^1$H dimension from experimental 2D $^{13}$C-$^1$H HSQC of mouse urine sample. (A), (C), (E): Experimental and reconstructed spectra are depicted as black lines and red dots, respectively. Deconvoluted individual peaks are depicted as blue lines. (B), (D), (F): The two HSQC traces and their sum are depicted as purple, cyan, and black lines, respectively. Pairs of panels (A, B), (C, D), and (E, F) show the same 1D spectral regions for comparison. The deconvolution by DEEP Picker1D was performed with model 1 with a PPP of 12.

It can be hard to assess the deconvolution accuracy of experimental spectra, since the ground truth, i.e. the individual isolated peaks and their parameters, are often unknown. For this reason, we constructed experimental spectra with overlaps from resolved spectra by co-adding

traces of a $^{13}$C-$^{1}$H HSQC spectrum of mouse urine along the direct $^{1}$H detection dimension at a fixed $^{13}$C chemical shift. Selected examples of overlapping peaks, both in isolation and as a superposition, are shown in **Figure 2**. The left column (Panels A, C, E) shows the experimental superpositions (black) together with their deconvolution (blue) and the full spectral reconstruction (red), which can be directly compared with the individual traces (purple and cyan)

in the right column (Panels B, D, F). **Figure 2A,B** shows two strongly overlapped peaks of different amplitude giving rise to a sum peak with a noticeable protrusion on its right flank, which are accurately deconvoluted and fitted by DEEP Picker1D and Voigt Fitter1D. **Figure 2C,D** shows a similar scenario, except that the amplitude ratio of the two peaks is around 35:1, which is much larger than in Panels A,B. Again, deconvolution was achieved with high accuracy.

**Figure 2E,F** demonstrate the deconvolution capacity for a challenging case of 4 moderately to strongly overlapped peaks. Although the peak at 3.139 ppm is wedged between two stronger peaks, it is successfully extracted by the peak picking and fitting algorithms. The final example (**Figure 3**) shows a region of the mouse urine spectrum (black) along with the deconvolution (blue) and reconstruction result. The algorithm deconvolutes the spectrum by identifying not

only the main peaks, but also all the minor peaks, including the peak at 7.809 ppm with confidence demonstrating the potential of the proposed deconvolution method in practice when encountering spectra with highly overlapped regions, such as those routinely collected for urine and other complex biofluids in the context of metabolomics.





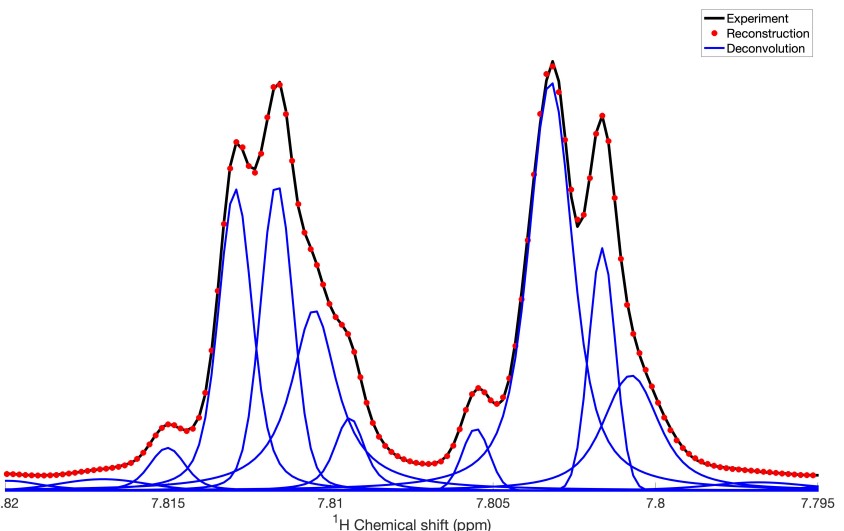

**Figure 3.** Application of DEEP Picker and Voigt Fitter 1D to a spectral region of 1D $^1$H spectrum of mouse urine. Experimental and reconstructed spectra are depicted as black lines and red dots, respectively. Deconvoluted individual peaks are depicted as blue lines. The deconvolution by DEEP Picker1D was performed with model 2 with a PPP of 8.

## 4. Discussion and Conclusion

In the vast majority of modern NMR applications, one of most critical steps in NMR spectral analysis is the identification of individual peaks along with their quantitative parametrization by lineshape fitting. The result of this procedure often dictates the usefulness, and ultimately the success, of the collected experiment. Traditional peak picking methods rely on clearly defined mathematical criteria, such as the properties of the 1$^{st}$ and 2$^{nd}$ derivative of the spectrum, to

identify individual peaks. These criteria are often too rigid to deal with spectral overlap scenarios encountered in practice. After proper training, a deep neural network like DEEP Picker1D, on the other hand, has a stunning ability to track major and minor spectral features surpassing the capacity of most human NMR practitioners. Through the combination of advanced machine-



learning by the convolutional deep neural network DEEP Picker1D and a peak fitting routine Voigt Fitter1D, it was demonstrated how 1D NMR spectral features of variable complexity can be deconvoluted into individual resonances in a reliable and accurate manner. The success rate of the method depends on the quality of spectra that can be affected by sample preparation, NMR data acquisition, and pre-processing. This concerns the elimination or suppression of the solvent signal or of a prominent background caused, for example, by the presence of a macromolecular matrix in the sample. Although apodization, zero-filling, phase and baseline correction are standard steps during data processing, they need to be applied judiciously to prevent suboptimal performance of spectral deconvolution and fitting. Phase errors of up to about $3^{\circ}$ can be tolerated but for larger phase distortions, DEEP Picker1D may interpret asymmetries in the peak shapes as shoulder peaks. Similarly, poor shimming of higher order shims, especially $z^2$ and $z^4$, can lead to systematic peak asymmetries across the spectrum, which DEEP Picker1D may interpret as shoulder peaks. In order to accurately recognize peak shapes DEEP Picker1D requires an adequate digital resolution, which is around 8 or 12 points across a single peak, depending also on the chosen DEEP Picker1D model. If needed, lower resolution spectra can be easily subjected to appropriated zero-filling to meet this criterion. Peak shapes should follow in good approximation Voigt profiles, which can be achieved by the application of common window functions as those described for the processing of the spectra in this work (cosine-squared and $2\pi$-Kaiser window functions, see Materials and Methods section). As discussed previously (Li et al., 2022a), the computational time of Voigt Fitter1D scales linearly with the number of peaks, allowing rapid fitting of complex 1D spectra with even thousands of peaks. The fitting of the 1D mouse urine spectrum with a total of 4500 Voigt shaped peaks took about 1 minute on a standard desktop computer. Like all nonlinear optimization software, Voigt Fitter1D cannot guarantee that

the final solution is the global $\chi^2$ minimum. Therefore, a nearly complete list of high-quality initial peaks returned by DEEP Picker1D that match the ground truth as closely as possible is key for the success of Voigt Fitter 1D.

A surge in metabolomics research over recent years has spurred the development of advanced quantitative tools for the analysis of complex NMR spectra both for 1D and 2D spectra. Some metabolomics software (Hao et al., 2014; Ravanbakhsh et al., 2015) are specifically geared toward the quantification of specific metabolites with known reference spectra, limiting their application to specific samples only, such as serum. In the case of DEEP Picker1D and Voigt

Fitter1D, the analysis is performed in a fully untargeted manner, i.e. without any molecular spectral templates, allowing its application to essentially any NMR spectrum that consists of resonances with Voigt lineshapes. The deconvolution results can then be further analyzed for example by querying against a spectral database or for quantitation of mixture component concentrations. In the case of a cohort of samples, the Voigt Fitter1D results can be used for

univariate or multivariate statistical analysis for the assessment of statistically significant differences between cohorts. The DEEP Picker1D and Voigt Fitter1D software can also be applied to a pseudo-2D series of 1D spectra for the extraction of longitudinal $R_1$, transverse $R_2$ relaxation parameters or translational diffusion constants by diffusion-ordered NMR (Johnson, 1999). The unique strength of the combination of DEEP Picker1D with Voigt Fitter1D is their

ability to accurately deconvolute and reconstruct NMR spectra of generic origin ranging from well-resolved to highly crowded, which should fulfill the growing needs in a wide range of contemporary NMR applications.

**Code availability**


The 1D version of DEEP Picker and Voigt Fitter are implemented in C/C++ and are now part of the DEEP Picker package, which is freely available from https://github.com/lidawei1975/deep under the GNU General Public License Agreement. They can also be accessed conveniently as a web server at https://spin.ccic.osu.edu/index.php/deep1d, which in addition provides an intuitive

interactive interface for the visual inspection of the results.

**Author contribution**

D.-W.L. and R.B. conceived and designed the project. D.-W.L. developed the machine learning framework and wrote the program. L.B.-L. prepared samples for NMR experiments. L.B.-L. and

A.L.H. performed NMR experiments. All authors contributed to model development and data analysis. D.-W.L. and R.B. prepared the manuscript.

**Competing interests**

The authors have a patent application pending.


**Financial support**

This work was supported by NIH (grant NIH R35GM139482) and NSF (grant MCB-2103637). All NMR experiments were performed at the shared CCIC NMR facility at the Ohio State University.



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
