# Peer review of "DEEP Picker1D and Voigt Fitter1D: A versatile tool set for the automated"

_Magnetic Resonance, 2022_

## Author Response (AR1)

**Reviewer #1**

This paper introduces a tool for the analysis of 1D NMR spectra of complex mixtures. It is based on a concept which has been recently published by the same authors in the case of 2D NMR. The method is based on machine learning and deep neural networks. It includes peak picking, fitting and deconvolution, and offers an impressive performance in the case of highly overlapped peaks. In particular, it seems very promising for metabolomics applications, and the code availability should allow a large community of users to test it on a variety of matrices (please do not make it a commercial tool!)

> Author response:
>
> Thank you for the comments. No worries, the DEEP Picker1D and Voigt Fitter1D software tools are freely available for academic users for download on github (https://github.com/lidawei1975/deep) under the GNU General Public License Agreement and as a public web server https://spin.ccic.osu.edu/index.php/deep1d

-page 6: in addition to the manufacturer-dependent pulse sequence names, please add a comprehensive name (such as "water suppression using excitation sculpting with a perfect echo")

> Author response:
>
> In the "NMR Experiments and Processing" section (p. 6, 7) the Bruker pulse sequences were annotated as requested by the reviewer. Specifically, "zgesgppe" is a $^1$H perfect-echo 1D experiment with excitation-sculpting water suppression and the "hsqcetgpsisp2.2" is a sensitivity-enhanced $^{13}$C-$^1$H HSQC with bi-level adiabatic decoupling.

-page 9: "Peak parameters, such as peak position, peak height and peak volume can then be directly used for downstream analysis, such as compound identification and quantitative NMR applications" --> this sentence gives the feeling that with the authors' tool, anyone can retrieve quantitative data from signals...this is only true if data have been acquired in quantitative conditions. This may sound obvious for most readers, but I would still suggest modifying the sentence as follows (or something similar): "Peak parameters, such as peak position, peak height and peak volume can then be directly used for downstream analysis, such as compound identification and quantitative NMR applications when incorporated into a quantitative NMR workflow"

> Author response:
>
> This is a valid point. We made this change on p. 9 as suggested by the reviewer.

-page 13: "It can be hard to assess the deconvolution accuracy of experimental spectra, since the ground truth, i.e. the individual isolated peaks and their parameters, are often unknown." --> in fact this is not so difficult, relying on synthetic mixtures of metabolites (for instance synthetic

urine or plasma) where concentrations are known. Such samples can be prepared in the lab with accurate gravimetric measurements, or even commercial. Since this is a methodological paper, I am not requesting the authors to perform such additional experiments, but that will be important for future papers relying on this tool (in particular when it comes to analytical chemistry considerations). Here I would just suggest to moderate the beginning of the paragraph by removing the first sentence: "To assess the deconvolution accuracy of our tool, we constructed experimental spectra with overlaps..."

> Author response:
>
> Thank you for the suggestion. We have modified this sentence as suggested on p.9.

-in the discussion, could the authors add a comment on how the method is expected to perform on a series of samples from a metabolomics cohort, and how it would account from peak shifting (due to pH variations, etc) across a metabolomics cohort?

> Author response:
>
> DEEP Picker1D and Voigt Fitter1D are general 1D NMR analysis tools that are applicable to a very wide range of 1D NMR spectra without requiring prior knowledge about the sample composition. Subsequent analysis and interpretation of the results returned by DP1D&VF1D will naturally depend on the specific application. In the case of metabolomics applications, peak shifts within cohorts of spectra are a well-known challenge and various approaches have been reported in the literature requiring additional knowledge about the molecular composition of the sample. Within our lab's on-going research program in metabolomics, we are currently developing an integrated workflow that includes metabolomics database query of cohorts of spectra using the output of DEEP Picker1D and Voigt Fitter1D. We have added the statement on p. 16:
>
> "Metabolomics query capabilities for the analysis of the output of DEEP Picker1D and Voigt Fitter1D, which will also take into account peak shifts caused, for example, by pH differences between samples is currently under development."

**Reviewer #2**

The procedure in this paper for esentially "peak picking" is quite good. My one comment is that while the peak fitting is very good, for the use in metabolomics one of the major issues is the identification of the peaks. They can have significatly different chemical shifts due to pH and ionic strenght differences. It would be good for the authors to suggest an approach to combinine their method with other methods using the AI approach for assignment of the fitted peaks.

> Author response:
>
> Thank you for the review. The final comment is similar to the last point of Reviewer #1. Peak annotation for the identification of known metabolites in complex metabolomics

mixtures requires the pairing of the DEEP Picker1D and Voigt Fitter1D output with a metabolomics database along with a strong query engine for the analysis of cohorts of spectra that takes peak shifts into account. Such work is currently in progress. We have added the following statement on p. 16:

"Metabolomics query capabilities for the analysis of the output of DEEP Picker1D and Voigt Fitter1D, which will also take into account peak shifts caused, for example, by pH differences between samples is currently under development."